# BAG OF TRICKS FOR ADVERSARIAL TRAINING

**Tianyu Pang, Xiao Yang, Yinpeng Dong, Hang Su, Jun Zhu**[*]
Department of Computer Science & Technology, Institute for AI, BNRist Center
Tsinghua-Bosch Joint ML Center, THBI Lab, Tsinghua University, Beijing, 100084 China
{pty17,yangxiao19,dyp17}@mails.tsinghua.edu.cn, {suhangss,dcszj}@tsinghua.edu.cn

## ABSTRACT

Adversarial training (AT) is one of the most effective strategies for promoting model robustness. However, recent benchmarks show that most of the proposed improvements on AT are less effective than simply early stopping the training procedure. This counter-intuitive fact motivates us to investigate the implementation details of tens of AT methods. Surprisingly, we find that the basic settings (e.g., weight decay, training schedule, etc.) used in these methods are highly inconsistent. In this work, we provide comprehensive evaluations on CIFAR-10, focusing on the effects of *mostly overlooked* training tricks and hyperparameters for adversarially trained models. Our empirical observations suggest that adversarial robustness is much more sensitive to some basic training settings than we thought. For example, a slightly different value of weight decay can reduce the model robust accuracy by more than 7%, which is probable to override the potential promotion induced by the proposed methods. We conclude a baseline training setting and re-implement previous defenses to achieve new state-of-the-art results[1]. These facts also appeal to more concerns on the overlooked confounders when benchmarking defenses.

## 1 INTRODUCTION

Adversarial training (AT) has been one of the most effective defense strategies against adversarial attacks (Biggio et al., 2013; Szegedy et al., 2014; Goodfellow et al., 2015). Based on the primary AT frameworks like PGD-AT (Madry et al., 2018), many improvements have been proposed from different perspectives, and demonstrate promising results (detailed in Sec. 2). However, the recent benchmarks (Croce & Hein, 2020b; Chen & Gu, 2020) find that simply early stopping the training procedure of PGD-AT (Rice et al., 2020) can attain the gains from almost all the previously proposed improvements, including the state-of-the-art TRADES (Zhang et al., 2019b).

This fact is somewhat striking since TRADES also executes early stopping (one epoch after decaying the learning rate) in their code implementation. Besides, the reported robustness of PGD-AT in Rice et al. (2020) is much higher than in Madry et al. (2018), even without early-stopping. This paradox motivates us to check the implementation details of these seminal works. We find that TRADES uses weight decay of $2 \times 10^{-4}$, Gaussian PGD initialization as $\delta_0 \sim \mathcal{N}(0, \alpha I)$, and eval mode of batch normalization (BN) when crafting adversarial examples, while Rice et al. (2020) use weight decay of $5 \times 10^{-4}$, uniform PGD initialization as $\delta_0 \sim \mathcal{U}(-\epsilon, \epsilon)$, and train mode of BN to generate adversarial examples. In our experiments on CIFAR-10 (e.g., Table 8), the two slightly different settings can differ the robust accuracy by $\sim 5\%$, which is significant according to the reported benchmarks.

To have a comprehensive study, we further investigate the implementation details of tens of papers working on the AT methods, some of which are summarized in Table 1. We find that even using the same model architectures, the basic hyperparameter settings (e.g., weight decay, learning rate schedule, etc.) used in these papers are highly inconsistent and customized, which could affect the model performance and may override the gains from the methods themselves. Under this situation, if we directly benchmark these methods using their released code or checkpoints, some actually effective improvements would be under-estimated due to the improper hyperparameter settings.

**Our contributions.** We evaluate the effects of a wide range of basic training tricks (e.g., warmup, early stopping, weight decay, batch size, BN mode, etc.) on the adversarially trained models. Our empirical results suggest that improper training settings can largely degenerate the model performance,

---

[*]Corresponding author.
[1]Code is available at **https://github.com/P2333/Bag-of-Tricks-for-AT**

Table 1: Hyperparameter settings and tricks used to implement different AT methods on CIFAR-10. We convert the training steps into epochs, and provide code links for reference in Table 11. Compared to the model architectures, the listed settings are easy to be neglected and paid less attention to unify.

| Method | l.r. | Total epoch (l.r. decay) | Batch size | Weight decay | Early stop (train / attack) | Warm-up (l.r. / pertub.) |
|---|---|---|---|---|---|---|
| Madry et al. (2018) | 0.1 | 200 (100, 150) | 128 | $2 \times 10^{-4}$ | No / No | No / No |
| Cai et al. (2018) | 0.1 | 300 (150, 250) | 200 | $5 \times 10^{-4}$ | No / No | No / Yes |
| Zhang et al. (2019b) | 0.1 | 76 (75) | 128 | $2 \times 10^{-4}$ | Yes / No | No / No |
| Wang et al. (2019) | 0.01 | 120 (60, 100) | 128 | $1 \times 10^{-4}$ | No / Yes | No / No |
| Qin et al. (2019) | 0.1 | 110 (100, 105) | 256 | $2 \times 10^{-4}$ | No / No | No / Yes |
| Mao et al. (2019) | 0.1 | 80 (50, 60) | 50 | $2 \times 10^{-4}$ | No / No | No / No |
| Carmon et al. (2019) | 0.1 | 100 (cosine anneal) | 256 | $5 \times 10^{-4}$ | No / No | No / No |
| Alayrac et al. (2019) | 0.2 | 64 (38, 46, 51) | 128 | $5 \times 10^{-4}$ | No / No | No / No |
| Shafahi et al. (2019b) | 0.1 | 200 (100, 150) | 128 | $2 \times 10^{-4}$ | No / No | No / No |
| Zhang et al. (2019a) | 0.05 | 105 (79, 90, 100) | 256 | $5 \times 10^{-4}$ | No / No | No / No |
| Zhang & Wang (2019) | 0.1 | 200 (60, 90) | 60 | $2 \times 10^{-4}$ | No / No | No / No |
| Atzmon et al. (2019) | 0.01 | 100 (50) | 32 | $1 \times 10^{-4}$ | No / No | No / No |
| Wong et al. (2020) | 0∼0.2 | 30 (one cycle) | 128 | $5 \times 10^{-4}$ | No / No | Yes / No |
| Rice et al. (2020) | 0.1 | 200 (100, 150) | 128 | $5 \times 10^{-4}$ | Yes / No | No / No |
| Ding et al. (2020) | 0.3 | 128 (51, 77, 102) | 128 | $2 \times 10^{-4}$ | No / No | No / No |
| Pang et al. (2020a) | 0.01 | 200 (100, 150) | 50 | $1 \times 10^{-4}$ | No / No | No / No |
| Zhang et al. (2020) | 0.1 | 120 (60, 90, 110) | 128 | $2 \times 10^{-4}$ | No / Yes | No / No |
| Huang et al. (2020) | 0.1 | 200 (cosine anneal) | 256 | $5 \times 10^{-4}$ | No / No | Yes / No |
| Cheng et al. (2020) | 0.1 | 200 (80, 140, 180) | 128 | $5 \times 10^{-4}$ | No / No | No / No |
| Lee et al. (2020) | 0.1 | 200 (100, 150) | 128 | $2 \times 10^{-4}$ | No / No | No / No |
| Xu et al. (2020) | 0.1 | 120 (60, 90) | 256 | $1 \times 10^{-4}$ | No / No | No / No |

while this degeneration may be mistakenly ascribed to the methods themselves. We provide a baseline recipe for PGD-AT on CIFAR-10 as an example, and demonstrate the generality of the recipe on training other frameworks like TRADES. As seen in Table 16, the retrained TRADES achieve new state-of-the-art performance on the AutoAttack benchmark (Croce & Hein, 2020b).

Although our empirical conclusions may not generalize to other datasets or tasks, we reveal the facts that adversarially trained models could be sensitive to certain training settings, which are usually neglected in previous work. These results also encourage the community to re-implement the previously proposed defenses with fine-tuned training settings to better explore their potentials.

## 2 RELATED WORK

In this section, we introduce related work on the adversarial defenses and recent benchmarks. We detail on the adversarial attacks in Appendix A.1.

### 2.1 ADVERSARIAL DEFENSES

To alleviate the adversarial vulnerability of deep learning models, many defense strategies have been proposed, but most of them can eventually be evaded by adaptive attacks (Carlini & Wagner, 2017b; Athalye et al., 2018). Other more theoretically guaranteed routines include training provably robust networks (Dvijotham et al., 2018a;b; Hein & Andriushchenko, 2017; Wong & Kolter, 2018) and obtaining certified models via randomized smoothing (Cohen et al., 2019). While these methods are promising, they currently do not match the state-of-the-art robustness under empirical evaluations.

The idea of adversarial training (AT) stems from the seminal work of Goodfellow et al. (2015), while other AT frameworks like PGD-AT (Madry et al., 2018) and TRADES (Zhang et al., 2019b) occupied the winner solutions in the adversarial competitions (Kurakin et al., 2018; Brendel et al., 2020). Based on these primary AT frameworks, many improvements have been proposed via encoding the mechanisms inspired from other domains, including ensemble learning (Tramèr et al., 2018; Pang et al., 2019), metric learning (Mao et al., 2019; Li et al., 2019; Pang et al., 2020c), generative modeling (Jiang et al., 2018; Pang et al., 2018b; Wang & Yu, 2019; Deng et al., 2020), semi-supervised learning (Carmon et al., 2019; Alayrac et al., 2019; Zhai et al., 2019), and self-supervised

Table 2: Test accuracy (%) under different **early stopping** and **warmup** on CIFAR-10. The model is ResNet-18 (results on WRN-34-10 is in Table 14). For early stopping attack iter., we denote, e.g., 40 / 70 as the epochs to increase the tolerance step by one (Zhang et al., 2020). For warmup, the learning rate and the maximal perturbation linearly increase from zero to preset values in 10 / 15 / 20 epochs.

| | Base | Early stopping attack iter. | | | Warmup on l.r. | | | Warmup on perturb. | | |
|---|---|---|---|---|---|---|---|---|---|---|
| | | 40 / 70 | 40 / 100 | 60 / 100 | 10 | 15 | 20 | 10 | 15 | 20 |
| Clean | 82.52 | 86.52 | 86.56 | 85.67 | 82.45 | 82.64 | 82.31 | 82.64 | 82.75 | 82.78 |
| PGD-10 | 53.58 | 52.65 | 53.22 | 52.90 | 53.43 | 53.29 | 53.35 | 53.65 | 53.27 | 53.62 |
| AA | 48.51 | 46.6 | 46.04 | 45.96 | 48.26 | 48.12 | 48.37 | 48.44 | 48.17 | 48.48 |

learning (Hendrycks et al., 2019; Chen et al., 2020a;b; Naseer et al., 2020). On the other hand, due to the high computational cost of AT, many efforts are devoted to accelerating the training procedure via reusing the computations (Shafahi et al., 2019b; Zhang et al., 2019a), adaptive adversarial steps (Wang et al., 2019; Zhang et al., 2020) or one-step training (Wong et al., 2020; Liu et al., 2020; Vivek B & Venkatesh Babu, 2020). The following works try to solve the side effects (e.g., catastrophic overfitting) caused by these fast AT methods (Andriushchenko & Flammarion, 2020; Li et al., 2020).

## 2.2 ADVERSARIAL BENCHMARKS

Due to the large number of proposed defenses, several benchmarks have been developed to rank the adversarial robustness of existing methods. Dong et al. (2020) perform large-scale experiments to generate robustness curves, which are used for evaluating typical defenses. Croce & Hein (2020b) propose AutoAttack, which is an ensemble of four selected attacks. They apply AutoAttack on tens of previous defenses and provide a comprehensive leader board. Chen & Gu (2020) propose the black-box RayS attack, and establish a similar leader board for defenses. In this paper, we mainly apply PGD attack and AutoAttack as two common ways to evaluate the models.

Except for the adversarial robustness, there are other efforts that introduce augmented datasets for accessing the robustness against general corruptions or perturbations. Mu & Gilmer (2019) introduce MNIST-C with a suite of 15 corruptions applied to the MNIST test set, while Hendrycks & Dietterich (2019) introduce ImageNet-C and ImageNet-P with common corruptions and perturbations on natural images. Evaluating robustness on these datasets can reflect the generality of the proposed defenses, and avoid overfitting to certain attacking patterns (Engstrom et al., 2019; Tramèr & Boneh, 2019).

## 3 BAG OF TRICKS

Our overarching goal is to investigate how the *usually overlooked* implementation details affect the performance of the adversarially trained models. Our experiments are done on CIFAR-10 (Krizhevsky & Hinton, 2009) under the $\ell_\infty$ threat model of maximal perturbation $\epsilon = 8/255$, without accessibility to additional data. We evaluate the models under 10-steps PGD attack (**PGD-10**) (Madry et al., 2018) and AutoAttack (**AA**) (Croce & Hein, 2020b). For the PGD attack, we apply untargeted mode using ground truth labels, step size of $2/255$, and 5 restarts for evaluation / no restart for training. For the AutoAttack[2], we apply the standard version, with no restart for AutoPGD and FAB, compared to 5 restarts for plus version. We consider some basic training tricks and perform ablation studies on each of them, based on the default training setting as described below:

**Default setting.** Following Rice et al. (2020), in the default setting, we apply the primary PGD-AT framework and the hyperparameters including batch size 128; SGD momentum optimizer with the initial learning rate of 0.1; weight decay $5 \times 10^{-4}$; ReLU activation function and no label smoothing; train mode for batch normalization when crafting adversarial examples. All the models are trained for 110 epochs with the learning rate decaying by a factor of 0.1 at 100 and 105 epochs, respectively. *We report the results on the checkpoint with the best PGD-10 accuracy.*

Note that our empirical observations and conclusions may not always generalize to other datasets or AT frameworks, but we emphasize the importance of using consistent implementation details (not only the same model architectures) to enable fair comparisons among different AT methods.

---

[2]https://github.com/fra31/auto-attack

Table 3: Test accuracy (%) under different **batch size** and **learning rate** (l.r.) on CIFAR-10. The basic l.r. is 0.1, while the scaled l.r. is, e.g., 0.2 for batch size 256, and 0.05 for batch size 64.

Table 4: Test accuracy (%) under different degrees of **label smoothing** (LS) on CIFAR-10. More evaluation results under, e.g., PGD-1000 can be found in Table 17.

| ResNet-18 | | | | |
|---|---|---|---|---|
| Batch | Basic l.r. | | Scaled l.r. | |
| size | Clean | PGD-10 | Clean | PGD-10 |
| 64 | 80.08 | 51.31 | 82.44 | 52.48 |
| 128 | 82.52 | **53.58** | - | - |
| 256 | 83.33 | 52.20 | 82.24 | 52.52 |
| 512 | 83.40 | 50.69 | 82.16 | 53.36 |
| **WRN-34-10** | | | | |
| Batch | Basic l.r. | | Scaled l.r. | |
| size | Clean | PGD-10 | Clean | PGD-10 |
| 64 | 84.20 | 54.69 | 85.40 | 54.86 |
| 128 | 86.07 | **56.60** | - | - |
| 256 | 86.21 | 52.90 | 85.89 | 56.09 |
| 512 | 86.29 | 50.17 | 86.47 | 55.49 |

| ResNet-18 | | | | |
|---|---|---|---|---|
| LS | Clean | PGD-10 | AA | RayS |
| 0 | 82.52 | 53.58 | 48.51 | 53.34 |
| 0.1 | 82.69 | 54.04 | 48.76 | 53.71 |
| 0.2 | 82.73 | 54.22 | 49.20 | 53.66 |
| 0.3 | 82.51 | 54.34 | **49.24** | 53.59 |
| 0.4 | 82.39 | 54.13 | 48.83 | 53.40 |
| **WRN-34-10** | | | | |
| LS | Clean | PGD-10 | AA | RayS |
| 0 | 86.07 | 56.60 | 52.19 | 60.07 |
| 0.1 | 85.96 | 56.88 | 52.74 | 59.99 |
| 0.2 | 86.09 | 57.31 | **53.00** | 60.28 |
| 0.3 | 85.99 | 57.55 | 52.70 | 61.00 |
| 0.4 | 86.19 | 57.63 | 52.71 | 60.64 |

Table 5: Test accuracy (%) using different **optimizers** on CIFAR-10. The model is ResNet-18 (results on WRN-34-10 is in Table 15). The initial learning rate for Adam and AdamW is 0.0001.

| | Mom | Nesterov | Adam | AdamW | SGD-GC | SGD-GCC |
|---|---|---|---|---|---|---|
| Clean | 82.52 | 82.83 | 83.20 | 81.68 | 82.77 | 82.93 |
| PGD-10 | 53.58 | 53.78 | 48.87 | 46.58 | 53.62 | 53.40 |
| AA | 48.51 | 48.22 | 44.04 | 42.39 | 48.33 | 48.51 |

## 3.1 EARLY STOPPING AND WARMUP

**Early stopping training epoch.** The trick of early stopping w.r.t. the training epoch was first applied in the implementation of TRADES (Zhang et al., 2019b), where the learning rate decays at the 75th epoch and the training is stopped at the 76th epoch. Later Rice et al. (2020) provide a comprehensive study on the overfitting phenomenon in AT, and advocate early stopping the training epoch as a general strategy for preventing adversarial overfitting, which could be triggered according to the PGD accuracy on a split validation set. Due to its effectiveness, we regard this trick as a default choice.

**Early stopping adversarial intensity.** Another level of early stopping happens on the adversarial intensity, e.g., early stopping PGD steps when crafting adversarial examples for training. This trick was first applied by the runner-up of the defense track in NeurIPS 2018 adversarial vision challenge (Brendel et al., 2020). Later efforts are devoted to formalizing this early stopping mechanism with different trigger rules (Wang et al., 2019; Zhang et al., 2020). Balaji et al. (2019) early stop the adversarial perturbation, which has a similar effect on the adversarial intensity. In the left part of Table 2, we evaluate the method proposed by Zhang et al. (2020) due to its simplicity. As seen, this kind of early stopping can improve the performance on clean data while keeping comparable accuracy under PGD-10. However, the performance under the stronger AutoAttack is degraded.

**Warmup w.r.t. learning rate.** Warmup w.r.t. learning rate is a general trick for training deep learning models (Goodfellow et al., 2016). In the adversarial setting, Wong et al. (2020) show that the one cycle learning rate schedule is one of the critical ingredients for the success of FastAT. Thus, we evaluate the effect of this trick for the piecewise learning rate schedule and PGD-AT framework. We linearly increase the learning rate from zero to the preset value in the first 10 / 15 / 20 epochs. As shown in the middle part of Table 2, the effect of warming up learning rate is marginal.

**Warmup w.r.t. adversarial intensity.** In the AT procedure, warmup can also be executed w.r.t. the adversarial intensity. Cai et al. (2018) propose the curriculum AT process to gradually increase the adversarial intensity and monitor the overfitting trend. Qin et al. (2019) increase the maximal

Table 6: Test accuracy (%) under different **non-linear activation function** on CIFAR-10. The model is ResNet-18. We apply the hyperparameters recommended by Xie et al. (2020) on ImageNet for the activation function. Here the notation ‡ indicates using weight decay of $5 \times 10^{-5}$, where applying weight decay of $5 \times 10^{-4}$ with these activations will lead to much worse model performance.

|  | ReLU | Leaky. | ELU ‡ | CELU ‡ | SELU ‡ | GELU | Softplus | Tanh ‡ |
|---|---|---|---|---|---|---|---|---|
| Clean | 82.52 | 82.11 | 82.17 | 81.37 | 78.88 | 80.42 | **82.80** | 80.13 |
| PGD-10 | 53.58 | 53.25 | 52.08 | 51.37 | 49.53 | 52.21 | **54.30** | 49.12 |

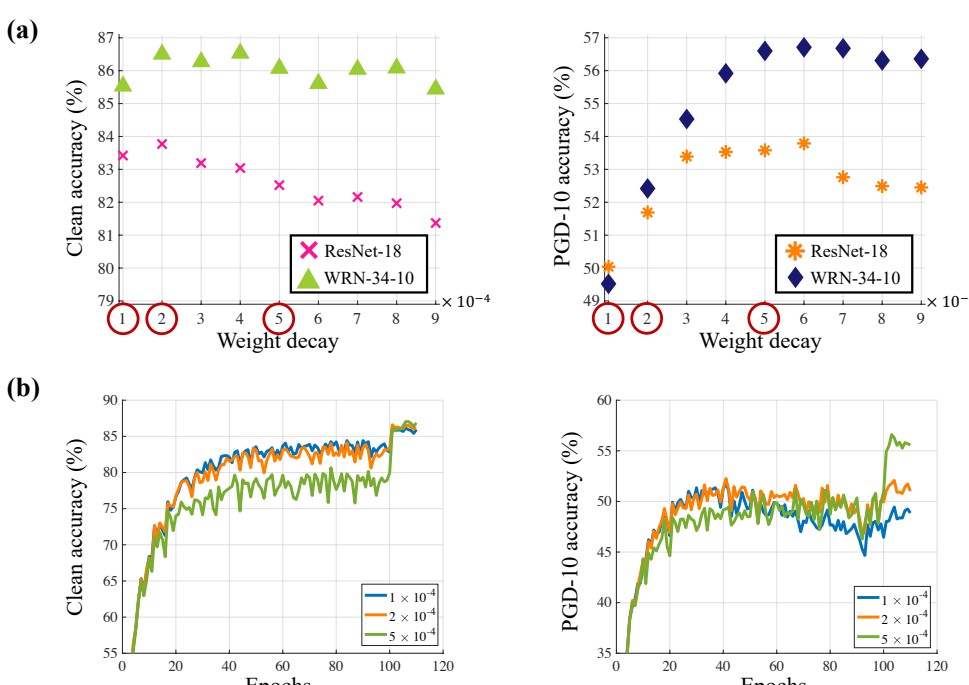

Figure 1: **(a)** Test accuracy w.r.t. different values of **weight decay**. The reported checkpoints correspond to the best PGD-10 accuracy (Rice et al., 2020). We test on two model architectures, and highlight (with red circles) three most commonly used weight decays in previous work; **(b)** Curves of test accuracy w.r.t. training epochs, where the model is WRN-34-10. We set weight decay be $1 \times 10^{-4}$, $2 \times 10^{-4}$, and $5 \times 10^{-4}$, respectively. We can observe that smaller weight decay can learn faster but also more tend to overfit w.r.t. the robust accuracy. In Fig. 4, we early decay the learning rate before the models overfitting, but weight decay of $5 \times 10^{-4}$ still achieve better robustness.

perturbation $\epsilon$ from zero to $8/255$ in the first 15 epochs. In the right part of Table 2, we linearly increase the maximal perturbation in the first 10 / 15 / 20 epochs, while the effect is still limited.

## 3.2 TRAINING HYPERPARAMETERS

**Batch size.** On the large-scale datasets like ImageNet (Deng et al., 2009), it has been recognized that the mini-batch size is an important factor influencing the model performance (Goyal et al., 2017), where larger batch size traverses the dataset faster but requires more memory usage. In the adversarial setting, Xie et al. (2019) use a batch size of 4096 to train a robust model on ImageNet, which achieves state-of-the-art performance under adversarial attacks. As to the defenses reported on the CIFAR-10 dataset, the mini-batch sizes are usually chosen between 128 and 256, as shown in Table 1. To evaluate the effect, we test on two model architectures and four values of batch size in Table 3. Since the number of training epochs is fixed to 110, we also consider applying the linear scaling rule introduced in Goyal et al. (2017), i.e., when the mini-batch size is multiplied by $k$, multiply the learning rate by $k$. We treat the batch size of 128 and the learning rate of 0.1 as a basic setting to obtain the factor $k$. We can observe that the batch size of 128 works well on CIFAR-10, while the linear scaling rule can benefit the cases with other batch sizes.

Table 7: Test accuracy (%) under different **BN modes** on CIFAR-10. We evaluate across several model architectures, since the BN layers have different positions in different models.

|  | BN mode | \multicolumn{6}{c}{Model architecture} | | | | | |
|---|---|---|---|---|---|---|---|
|  |  | ResNet-18 | SENet-18 | DenseNet-121 | GoogleNet | DPN26 | WRN-34-10 |
| Clean | train | 82.52 | 82.20 | 85.38 | 83.97 | 83.67 | 86.07 |
|  | eval | 83.48 | 84.11 | 86.33 | 85.26 | 84.56 | 87.38 |
|  | - | +0.96 | +1.91 | +0.95 | +1.29 | +0.89 | +1.31 |
| PGD-10 | train | 53.58 | 54.01 | 56.22 | 53.76 | 53.88 | 56.60 |
|  | eval | 53.64 | 53.90 | 56.11 | 53.77 | 53.41 | 56.04 |
|  | - | +0.06 | -0.11 | -0.11 | +0.01 | -0.47 | -0.56 |
| AA | train | 48.51 | 48.72 | 51.58 | 48.73 | 48.50 | 52.19 |
|  | eval | 48.75 | 48.95 | 51.24 | 48.83 | 48.30 | 51.93 |
|  | - | +0.24 | +0.23 | -0.34 | +0.10 | -0.20 | -0.26 |

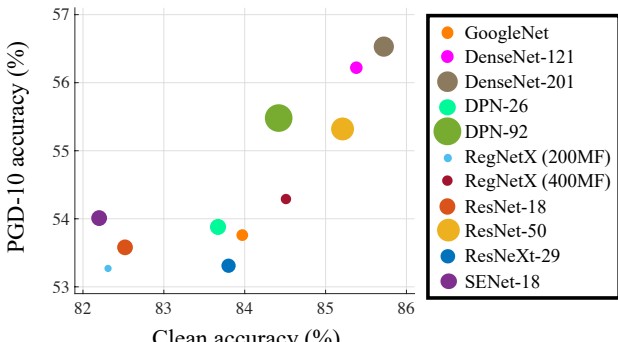

Figure 2: Clean accuracy vs. PGD-10 accuracy for different **model architectures**. The circle sizes are proportional to the number of parameters that specified in Table 12.

**Label smoothing (LS).** Shafahi et al. (2019a) propose to utilize LS to mimic adversarial training. Pang et al. (2019) also find that imposing LS on the ensemble prediction can alleviate the adversarial transferability among individual members. Unfortunately, combing LS with standard training cannot prevent the models from evaded by adaptive attacks (Tramer et al., 2020) or larger iteration steps (Summers & Dinneen, 2018). Beyond previous observations, we further evaluate the effect of LS on adversarial training. As shown in Table 4 and Table 17, mild LS can improve $0.5 \sim 1\%$ robust accuracy under the strong attacks we evaluated, including AutoAttack and PGD-1000, without affecting the clean performance. This can be regarded as the effect induced by calibrating the confidence (Stutz et al., 2020) of adversarially trained models ($80\% \sim 85\%$ accuracy on clean data). In contrast, excessive LS could degrade the robustness (e.g., LS = 0.3 vs. LS = 0.4 on ResNet-18), which is consistent with the recent observations in Jiang et al. (2020) (they use LS = 0.5). However, since LS is known for its potential gradient masking effect, we advocate careful evaluations when applying this trick on the proposed defenses, following the suggestions in Carlini et al. (2019).

**Optimizer.** Most of the AT methods apply SGD with momentum as the optimizer. The momentum factor is usually set to be 0.9 with zero dampening. In other cases, Carmon et al. (2019) apply SGD with Nesterov, and Rice et al. (2020) apply Adam for cyclic learning rate schedule. We test some commonly used optimizers in Table 5, as well as the decoupled AdamW (Loshchilov & Hutter, 2019) and the recently proposed gradient centralization trick SGD-GC / SGD-GCC (Yong et al., 2020). We can find that SGD-based optimizers (e.g., Mom, Nesterov, SGD-GC / SGD-GCC) have similar performance, while Adam / AdamW performs worse for piecewise learning rate schedule.

**Weight decay.** As observed in Table 1, three different values of weight decay are used in previous defenses, including $1 \times 10^{-4}$, $2 \times 10^{-4}$, and $5 \times 10^{-4}$. While $5 \times 10^{-4}$ is a fairly widely used value for weight decay in deep learning, the prevalence of the value $2 \times 10^{-4}$ should stem from Madry et al. (2018) in the adversarial setting. In Fig. 1(a), we report the best test accuracy under different values of weight decay[3]. We can see that the gap of robust accuracy can be significant due to slightly

---

[3]Note that Rice et al. (2020) also investigate the effect of different weight decay (i.e., $\ell_2$ regularization), but they focus on a coarse value range of $\{5 \times 10^k\}$, where $k \in \{-4, -3, -2, -1, 0\}$.

Table 8: The default hyperparameters include batch size 128 and SGD momentum optimizer. The AT framework is **PGD-AT**. We highlight the setting used by the implementation in Rice et al. (2020).

| Architecture | Label smooth | Weight decay | Activation function | BN mode | Accuracy | | |
|---|---|---|---|---|---|---|---|
| | | | | | Clean | PGD-10 | AA |
| WRN-34-10 | 0 | $1 \times 10^{-4}$ | ReLU | train | 85.87 | 49.45 | 46.43 |
| | 0 | $2 \times 10^{-4}$ | ReLU | train | 86.14 | 52.08 | 48.72 |
| | 0 | $5 \times 10^{-4}$ | ReLU | train | 86.07 | 56.60 | 52.19 |
| | 0 | $5 \times 10^{-4}$ | ReLU | eval | **87.38** | 56.04 | 51.93 |
| | 0 | $5 \times 10^{-4}$ | Softplus | train | 86.60 | 56.44 | 52.70 |
| | 0.1 | $5 \times 10^{-4}$ | Softplus | train | 86.42 | 57.22 | **53.01** |
| | 0.1 | $5 \times 10^{-4}$ | Softplus | eval | 86.34 | 56.38 | 52.21 |
| | 0.2 | $5 \times 10^{-4}$ | Softplus | train | 86.10 | 56.55 | 52.91 |
| | 0.2 | $5 \times 10^{-4}$ | Softplus | eval | 86.98 | 56.21 | 52.10 |
| WRN-34-20 | 0 | $1 \times 10^{-4}$ | ReLU | train | 86.21 | 49.74 | 47.58 |
| | 0 | $2 \times 10^{-4}$ | ReLU | train | 86.73 | 51.39 | 49.03 |
| | 0 | $5 \times 10^{-4}$ | ReLU | train | 86.97 | 57.57 | 53.26 |
| | 0 | $5 \times 10^{-4}$ | ReLU | eval | 87.62 | 57.04 | 53.14 |
| | 0 | $5 \times 10^{-4}$ | Softplus | train | 85.80 | 57.84 | 53.64 |
| | 0.1 | $5 \times 10^{-4}$ | Softplus | train | 85.69 | 57.86 | **53.66** |
| | 0.1 | $5 \times 10^{-4}$ | Softplus | eval | **87.86** | 57.33 | 53.23 |
| | 0.2 | $5 \times 10^{-4}$ | Softplus | train | 84.82 | 57.93 | 53.39 |
| | 0.2 | $5 \times 10^{-4}$ | Softplus | eval | 87.58 | 57.19 | 53.26 |

Region of correct predictions  Region of wrong predictions

Figure 3: Random normal cross-sections of the decision boundary for PGD-AT with different **weight decay**. The model architecture is WRN-34-10. Following the examples in Moosavi-Dezfooli et al. (2019), we craft PGD-10 perturbation as the normal direction $v$, and $r$ be a random direction, under the $\ell_\infty$ constraint of $8/255$. The values of x-axis and y-axis represent the multiplied scale factors.

different values of weight decay (e.g., up to $\sim 7\%$ for $1 \times 10^{-4}$ vs. $5 \times 10^{-4}$). Besides, in Fig. 1(b) we plot the learning curves of test accuracy w.r.t. training epochs. Note that smaller values of weight decay make the model learn faster in the initial phase, but the overfitting phenomenon also appears earlier. In Fig. 3, we visualize the cross sections of the decision boundary. We can see that proper values of weight decay (e.g., $5 \times 10^{-4}$) can enlarge margins from decision boundary and improve robustness. Nevertheless, as shown in the left two columns, this effect is less significant on promoting clean accuracy. As a result, weight decay is a critical and usually neglected ingredient that largely influences the robust accuracy of adversarially trained models. In contrast, the clean accuracy is much less sensitive to weight decay, for both adversarially and standardly (shown in Fig. 5) trained models.

**Activation function.** Most of the previous AT methods apply ReLU as the non-linear activation function in their models, while Xie et al. (2020) empirically demonstrate that smooth activation functions can better improve model robustness on ImageNet. Following their settings, we test if a similar conclusion holds on CIFAR-10. By comparing the results on ReLU and Softplus in Table 6 (for PGD-AT) and Table 13 (for TRADES), we confirm that smooth activation indeed benefits model robustness for ResNet-18. However, as shown in Table 8 (for PGD-AT) and Table 9 (for TRADES), this benefit is less significant on larger models like WRN. Thus we deduce that smaller model capacity can benefit more from the smoothness of activation function. Besides, as shown in Table 6, models trained on CIFAR-10 seem to prefer activation function $\sigma(x)$ with zero truncation, i.e., $\sigma(x) \geq 0$. Those with negative return values like ELU, LeakyReLU, Tanh have worse performance than ReLU.

Table 9: Test accuracy (%). The AT framework is **TRADES**. We highlight the setting used by the original implementation in Zhang et al. (2019b). As listed in Table 16, our retrained TRADES models can achieve state-of-the-art performance in the AutoAttack benchmark.

| Architecture | Weight decay | BN mode | Activation | Clean | PGD-10 | AA |
|---|---|---|---|---|---|---|
| *Threat model: $\ell_\infty$ constraint, $\epsilon = 0.031$* | | | | | | |
| | $2 \times 10^{-4}$ | train | ReLU | 83.86 | 54.96 | 51.52 |
| | $2 \times 10^{-4}$ | eval | ReLU | 85.17 | 55.10 | 51.85 |
| WRN-34-10 | $5 \times 10^{-4}$ | train | ReLU | 84.17 | 57.34 | 53.51 |
| | $5 \times 10^{-4}$ | eval | ReLU | **85.34** | 58.54 | **54.64** |
| | $5 \times 10^{-4}$ | eval | Softplus | 84.66 | 58.05 | 54.20 |
| WRN-34-20 | $5 \times 10^{-4}$ | eval | ReLU | **86.93** | 57.93 | **54.42** |
| | $5 \times 10^{-4}$ | eval | Softplus | 85.43 | 57.94 | 54.32 |
| *Threat model: $\ell_\infty$ constraint, $\epsilon = 8/255$* | | | | | | |
| Architecture | Weight decay | BN mode | Activation | Clean | PGD-10 | AA |
| | $2 \times 10^{-4}$ | train | ReLU | 84.50 | 54.60 | 50.94 |
| | $2 \times 10^{-4}$ | eval | ReLU | 85.17 | 54.58 | 51.54 |
| WRN-34-10 | $5 \times 10^{-4}$ | train | ReLU | 84.04 | 57.41 | 53.83 |
| | $5 \times 10^{-4}$ | eval | ReLU | **85.48** | 57.45 | 53.80 |
| | $5 \times 10^{-4}$ | eval | Softplus | 84.24 | 57.59 | **53.88** |
| | $2 \times 10^{-4}$ | train | ReLU | 84.50 | 53.86 | 51.18 |
| | $2 \times 10^{-4}$ | eval | ReLU | 85.48 | 53.21 | 50.59 |
| WRN-34-20 | $5 \times 10^{-4}$ | train | ReLU | 85.87 | 57.40 | 54.22 |
| | $5 \times 10^{-4}$ | eval | ReLU | **86.43** | 57.91 | **54.39** |
| | $5 \times 10^{-4}$ | eval | Softplus | 85.51 | 57.50 | 54.21 |

**Model architecture.** Su et al. (2018) provide a comprehensive study on the robustness of standardly trained models, using different model architectures. For the adversarially trained models, it has been generally recognized that larger model capacity can usually lead to better robustness (Madry et al., 2018). Recently, Guo et al. (2020) blend in the technique of AutoML to explore robust architectures. In Fig. 2, we perform similar experiments on more hand-crafted model architectures. The selected models have comparable numbers of parameters. We can observe that DenseNet can achieve both the best clean and robust accuracy, while being memory-efficient (but may require longer inference time). This is consistent with the observation in Guo et al. (2020) that residual connections can benefit the AT procedure. Interestingly, Wu et al. (2020) demonstrate that residual connections allow easier generation of highly transferable adversarial examples, while in our case this weakness for the standardly trained models may turn out to strengthen the adversarially trained models.

**Batch normalization (BN) mode.** When crafting adversarial examples in the training procedure, Zhang et al. (2019b) use eval mode for BN, while Rice et al. (2020) and Madry et al. (2018) use train mode for BN. Since the parameters in the BN layers are not updated in this progress, the difference between these two modes is mainly on the recorded moving average BN mean and variance used in the test phase. As pointed out in Xie & Yuille (2020), properly dealing with BN layers is critical to obtain a well-performed adversarially trained model. Thus in Table 7, we employ the train or eval mode of BN for crafting adversarial examples during training, and report the results on different model architectures to dig out general rules. As seen, using eval mode for BN can increase clean accuracy, while keeping comparable robustness. We also advocate for the eval mode, because if we apply train mode for multi-step PGD attack, the BN mean and variance will be recorded for every intermediate step, which could blur the adversarial distribution used by BN layers during inference.

---

**Takeaways:**
**(i)** Slightly different values of weight decay could largely affect the robustness of trained models;
**(ii)** Moderate label smoothing and linear scaling rule on l.r. for different batch sizes are beneficial;
**(iii)** Applying eval BN mode to craft training adversarial examples can avoid blurring the distribution;
**(iv)** Early stopping the adversarial steps or perturbation may degenerate worst-case robustness;
**(v)** Smooth activation benefits more when the model capacity is not enough for adversarial training.

---

Table 10: Test accuracy (%). The considered AT frameworks are **FastAT** and **FreeAT**. The model architecture is WRN-34-10. Detailed settings used for these defenses are described in Sec. 3.5.

| Defense | Label smooth | Weight decay | BN mode | Accuracy | | |
|---|---|---|---|---|---|---|
| | | | | Clean | PGD-10 | AA |
| FastAT (Wong et al., 2020) | 0 | $2 \times 10^{-4}$ | train | 82.19 | 47.47 | 42.99 |
| | 0 | $5 \times 10^{-4}$ | train | 82.93 | 48.48 | 44.06 |
| | 0 | $5 \times 10^{-4}$ | eval | **84.00** | 48.16 | 43.66 |
| | 0.1 | $5 \times 10^{-4}$ | train | 82.83 | **48.76** | **44.50** |
| FreeAT (Shafahi et al., 2019b) | 0 | $2 \times 10^{-4}$ | train | 87.42 | 47.66 | 44.24 |
| | 0 | $5 \times 10^{-4}$ | train | 88.17 | 48.90 | 45.66 |
| | 0 | $5 \times 10^{-4}$ | eval | **88.26** | 48.50 | 45.49 |
| | 0.1 | $5 \times 10^{-4}$ | train | 88.07 | **49.26** | **45.91** |

## 3.3 COMBINATION OF TRICKS

In the above, we separately evaluate the effect of each training trick in the AT procedure. Now we investigate combining the selected useful tricks, which involve label smoothing, weight decay, activation function and BN mode. As demonstrated in Table 8, the improvements are not ideally additive by combining different tricks, while label smoothing and smooth activation function are helpful, but not significant, especially when we apply model architectures with a larger capacity.

We also find that the high performance of the models trained by Rice et al. (2020) partially comes from its reasonable training settings, compared to previous work. Based on these, we provide a trick list for training robust models on CIFAR-10 for reference.

> **Baseline setting (CIFAR-10):**
> Batch size 128; SGD momentum optimizer; weight decay $5 \times 10^{-4}$; eval mode BN for generating adversarial examples; warmups are not necessary; moderate label smoothing ($0.1 \sim 0.2$) and smooth activation function could be beneficial; model architecture with residual connections.

## 3.4 RE-IMPLEMENTATION OF TRADES

As a sanity check, we re-implement TRADES to see if our conclusions derived from PGD-AT can generalize and provide the results in Table 9. We can observe that after simply changing the weight decay from $2 \times 10^{-4}$ to $5 \times 10^{-4}$, the clean accuracy of TRADES improves by $\sim 1\%$ and the AA accuracy improves by $\sim 4\%$, which make the trained model surpass the previously state-of-the-art models reported by the AutoAttack benchmark, as listed in Table 16. This fact highlights the importance of employing a standardized training setting for fair comparisons of different AT methods.

## 3.5 EVALUATIONS ON OTHER AT FRAMEWORKS

To examine the universality of our observations on PGD-AT and TRADES, we further evaluate on other AT frameworks, including FastAT (Wong et al., 2020) and FreeAT (Shafahi et al., 2019b). We base on the FastAT code[4] to implement the methods. Specifically, for FastAT, we use cyclic learning rate schedule with $l_{\min} = 0$ and $l_{\max} = 0.2$, training for 15 epochs. For FreeAT, we also use cyclic learning rate schedule with $l_{\min} = 0$ and $l_{\max} = 0.04$, training for 24 epochs with mini-batch replays be 4. The results are provided in Table 10. We can find that our observations generalize well to other AT frameworks, which verifies that the proposed baseline setting could be a decent default choice for adversarial training on CIFAR-10.

## 4 CONCLUSION

In this work, we take a step in examining how the usually neglected implementation details impact the performance of adversarially trained models. Our empirical results suggest that compared to clean accuracy, robustness is more sensitive to some seemingly unimportant differences in training settings. Thus when building AT methods, we should more carefully fine-tune the training settings (on validation sets), or follow certain long-tested setup in the adversarial setting.

---

[4]https://github.com/locuslab/fast_adversarial

## ACKNOWLEDGEMENTS

This work was supported by the National Key Research and Development Program of China (Nos. 2020AAA0104304, 2017YFA0700904), NSFC Projects (Nos. 61620106010, 62076147, U19B2034, U19A2081), Beijing Academy of Artificial Intelligence (BAAI), Tsinghua-Huawei Joint Research Program, a grant from Tsinghua Institute for Guo Qiang, Tiangong Institute for Intelligent Computing, and the NVIDIA NVAIL Program with GPU/DGX Acceleration. Tianyu Pang was supported by MSRA Fellowship and Baidu Scholarship.

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

## A    TECHNICAL DETAILS

In this section we introduce more related backgrounds and technical details for reference.

### A.1    ADVERSARIAL ATTACKS

Since the seminal L-BFGS and FGSM attacks (Szegedy et al., 2014; Goodfellow et al., 2015), a large amount of attacking methods on generating adversarial examples have been introduced. In the white-box setting, gradient-based methods are popular and powerful, which span in the $\ell_\infty$ threat model (Nguyen et al., 2015; Madry et al., 2018), $\ell_2$ threat model (Carlini & Wagner, 2017a), $\ell_1$ threat model (Chen et al., 2018), and $\ell_0$ threat model (Papernot et al., 2016). In the black-box setting, the attack strategies are much more diverse. These include transfer-based attacks (Dong et al., 2018; 2019; Cheng et al., 2019b), quasi-gradient attacks (Chen et al., 2017a; Uesato et al., 2018; Ilyas et al., 2018), and decision-based attacks (Brendel et al., 2018; Cheng et al., 2019a). Adversarial attacks can be also realized in the physical world (Kurakin et al., 2017; Song et al., 2018a). Below we formulate the PGD attack and AutoAttack that we used in our evaluations.

**PGD attack.** One of the most commonly studied adversarial attack is the projected gradient descent (PGD) method (Madry et al., 2018). Let $x_0$ be a randomly perturbed sample in the neighborhood of the clean input $x$, then PGD iteratively crafts the adversarial example as

$$x_i = \text{clip}_{x,\epsilon}(x_{i-1} + \epsilon_i \cdot \text{sign}(\nabla_{x_{i-1}}\mathcal{L}(x_{i-1}, y))), \tag{1}$$

where $\text{clip}_{x,\epsilon}(\cdot)$ is the clipping function and $\mathcal{L}$ is the adversarial objective. The accuracy under PGD attack has been a standard metric to evaluate the model robustness.

**AutoAttack.** Croce & Hein (2020b) first propose the Auto-PGD (APGD) algorithm, where the main idea is to automatically tune the adversarial step sizes according to the optimization trend. As to the adversarial objective, except for the traditional cross-entropy (CE) loss, they develop a new difference of logits ratio (DLR) loss as

$$\text{DLR}(x, y) = -\frac{z_y - \max_{i \neq y} z_i}{z_{\pi_1} - z_{\pi_3}}, \tag{2}$$

where $z$ is the logits and $\pi$ is the ordering which sorts the components of $z$. Finally, the authors propose to group $\text{APGD}_{\text{CE}}$ and $\text{APGD}_{\text{DLR}}$ with FAB (Croce & Hein, 2020a) and square attack (Andriushchenko et al., 2020) to form the AutoAttack (AA).

### A.2    REFERENCE CODES

In Table 11, we provide the code links for the referred defenses. The summarized training settings are either described in their papers or manually retrieved by us in their code implementations.

Table 11: We summarize the code links for the referred defense methods in Table 1.

| Method | Code link |
|---|---|
| Madry et al. (2018) | github.com/MadryLab/cifar10_challenge |
| Cai et al. (2018) | github.com/sunblaze-ucb/curriculum-adversarial-training-CAT |
| Zhang et al. (2019b) | github.com/yaodongyu/TRADES |
| Wang et al. (2019) | github.com/YisenWang/dynamic_adv_training |
| Mao et al. (2019) | github.com/columbia/Metric_Learning_Adversarial_Robustness |
| Carmon et al. (2019) | github.com/yaircarmon/semisup-adv |
| Alayrac et al. (2019) | github.com/deepmind/deepmind-research/unsupervised_adversarial_training |
| Shafahi et al. (2019b) | github.com/ashafahi/free_adv_train |
| Zhang et al. (2019a) | github.com/a1600012888/YOPO-You-Only-Propagate-Once |
| Zhang & Wang (2019) | github.com/Haichao-Zhang/FeatureScatter |
| Atzmon et al. (2019) | github.com/matanatz/ControllingNeuralLevelsets |
| Wong et al. (2020) | github.com/locuslab/fast_adversarial |
| Rice et al. (2020) | github.com/locuslab/robust_overfitting |
| Ding et al. (2020) | github.com/BorealisAI/mma_training |
| Pang et al. (2020a) | github.com/P2333/Max-Mahalanobis-Training |
| Zhang et al. (2020) | github.com/zjfheart/Friendly-Adversarial-Training |
| Huang et al. (2020) | github.com/LayneH/self-adaptive-training |
| Lee et al. (2020) | github.com/Saehyung-Lee/cifar10_challenge |

## A.3 Model architectures

We select some typical hand-crafted model architectures as the objects of study, involving DenseNet (Huang et al., 2017), GoogleNet (Szegedy et al., 2015), (PreAct) ResNet (He et al., 2016), SENet (Hu et al., 2018), WRN (Zagoruyko & Komodakis, 2016), DPN (Chen et al., 2017b), ResNeXt (Xie et al., 2017), and RegNetX (Radosavovic et al., 2020). The models are implemented by `https://github.com/kuangliu/pytorch-cifar`.

Table 12: Number of parameters for different model architectures.

| Architecture | # of param. | Architecture | # of param. | Architecture | # of param. |
|---|---|---|---|---|---|
| DenseNet-121 | 28.29 M | DPN26 | 46.47 M | GoogleNet | 24.81 M |
| DenseNet-201 | 73.55 M | DPN92 | 137.50 M | ResNeXt-29 | 36.65 M |
| RegNetX (200MF) | 9.42 M | ResNet-18 | 44.70 M | SENet-18 | 45.09 M |
| RegNetX (400MF) | 19.34 M | ResNet-50 | 94.28 M | WRN-34-10 | 193.20 M |

## A.4 Inference-phase adversarial defenses

Except for enhancing the models in the training phase, there are other methods that intend to improve robustness in the inference phase. These attempts include performing local linear transformation like adding Gaussian noise (Tabacof & Valle, 2016), different operations of image processing (Guo et al., 2018; Xie et al., 2018; Raff et al., 2019) or specified inference principle (Pang et al., 2020b). On the other hand, detection-based methods aim to filter out adversarial examples and resort to higher-level intervention. Although detection is a suboptimal strategy compared to classification, it can avoid over-confident wrong decisions. These efforts include training auxiliary classifiers to detect adversarial inputs (Metzen et al., 2017), designing detection statistics (Feinman et al., 2017; Ma et al., 2018; Pang et al., 2018a), or basing on additional probabilistic models (Song et al., 2018b).

## A.5 Concurrent work

Gowal et al. (2020) also provide a comprehensive study on different training tricks of AT, and push forward the state-of-the-art performance of adversarially trained models on MNIST, CIFAR-10 and CIFAR-100. While they analyze some properties that we also analyze in this paper (such as training batch size, label smoothing, weight decay, activation functions), they also complement our analyses with experiments on, e.g., weight moving average and data quality. Both of our works reveal the importance of training details in the process of AT, and contribute to establishing more justified perspectives for evaluating AT methods.

# B ADDITIONAL RESULTS

In this section, we provide additional results to further support the conclusions in the main text.

## B.1 EARLY DECAYS LEARNING RATE

As shown in Fig. 1, smaller values of weight decay make the training faster but also more tend to overfit. So in Fig. 4, we early decay the learning rate at 40 and 45 epochs, rather than 100 and 105 epochs. We can see that the models can achieve the same clean accuracy, but the weight decay of $5 \times 10^{-4}$ can still achieve better robustness. Besides, in Fig. 5, we use different values of weight decay for standard training, where the models can also achieve similar clean accuracy. These results demonstrate that adversarial robustness is a more difficult target than clean performance, and is more sensitive to the training hyperparameters, both for standardly and adversarially trained models.

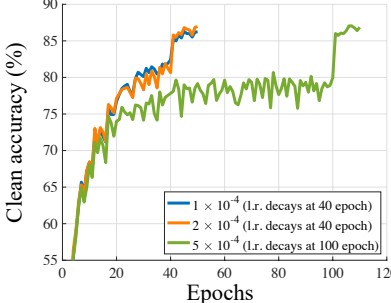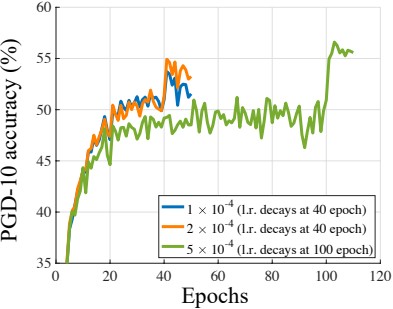

Figure 4: Curves of test accuracy w.r.t. training epochs, where the model is WRN-34-10. Here we early decay the learning rate at 40 and 45 epochs for the cases of weight decay $1 \times 10^{-4}$ and $2 \times 10^{-4}$, just before they overfitting. We can see that the models can achieve the same clean accuracy as weight decay $5 \times 10^{-4}$, but still worse robustness.

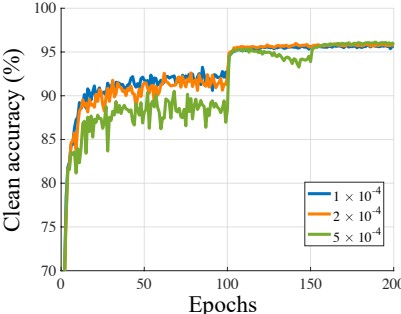

Figure 5: Curves of test accuracy w.r.t. training epochs. The model architecture is WRN-34-10, and is standardly trained on CIFAR-10. We can observe that the final performance of each model is comparable, which means that clean accuracy is less sensitive to different values of weight decay. This observation also holds for the adversarially trained models as shown in Fig. 1.

## B.2 THE EFFECT OF SMOOTH ACTIVATION FUNCTION

In Table 13 we test the effect of Softplus and BN mode on ResNet-18.

Table 13: Test accuracy (%) of **TRADES**. We compare with the results in Table 6 to check the effect of smooth activation function on TRADES, as well as the compatibility of it with eval BN mode.

| Architecture | Weight decay | BN mode | Activation | Clean | PGD-10 | AA |
|---|---|---|---|---|---|---|
| | *Threat model: $\ell_\infty$ constraint, $\epsilon = 8/255$* | | | | | |
| | $5 \times 10^{-4}$ | train | ReLU | 80.23 | 53.60 | 48.96 |
| | $5 \times 10^{-4}$ | train | Softplus | 81.26 | 54.58 | 50.35 |
| ResNet-18 | $5 \times 10^{-4}$ | eval | ReLU | 81.45 | 53.51 | 49.06 |
| | $5 \times 10^{-4}$ | eval | Softplus | 82.37 | 54.37 | 50.51 |

## B.3 RESULTS OF EARLY STOPPING, WARMUP, AND OPTIMIZERS ON WRN-34-10

In Table 14 and Table 15, we provide the results on WRN-34-10.

Table 14: Test accuracy (%) under different **early stopping** and **warmup** on CIFAR-10. The model is WRN-34-10. For early stopping attack iterations, we denote, e.g., 40 / 70 as the epochs to increase the tolerance step by one (Zhang et al., 2020). For warmup, the learning rate (l.r.) and the maximal perturbation (perturb.) linearly increase from zero to the preset value in the first 10 / 15 / 20 epochs.

| | Base | Early stopping attack iter. | | | Warmup on l.r. | | | Warmup on perturb. | | |
|---|---|---|---|---|---|---|---|---|---|---|
| | | 40 / 70 | 40 / 100 | 60 / 100 | 10 | 15 | 20 | 10 | 15 | 20 |
| Clean | 86.07 | 88.29 | 88.25 | 88.81 | 86.35 | 86.63 | 86.41 | 86.66 | 86.43 | 86.73 |
| PGD-10 | 56.60 | 56.06 | 55.49 | 56.41 | 56.31 | 56.60 | 56.28 | 56.25 | 56.37 | 55.65 |
| AA | 52.19 | 50.19 | 49.44 | 49.81 | 51.96 | 52.13 | 51.75 | 51.88 | 52.06 | 51.70 |

Table 15: Test accuracy (%) using different **optimizers** on CIFAR-10. The model is WRN-34-10. The initial learning rate for Adam and AdamW is 0.0001, while for other optimizers is 0.1.

| | Mom | Nesterov | Adam | AdamW | SGD-GC | SGD-GCC |
|---|---|---|---|---|---|---|
| Clean | 86.07 | 86.80 | 81.00 | 80.72 | 86.70 | 86.67 |
| PGD-10 | 56.60 | 56.34 | 52.54 | 50.32 | 56.06 | 56.14 |
| AA | 52.19 | 51.93 | 46.52 | 45.79 | 51.75 | 51.65 |

## B.4 RANK IN THE AUTOATTACK BENCHMARK

The models evaluated in this paper are all retrained based on the released codes (Zhang et al., 2019b; Rice et al., 2020). Now we compare our trained models with the AutoAttack public benchmark, where the results of previous work are based on the released pretrained models. In Table 16, we retrieve our results in Table 9 on the TRADES model where we simply change the weight decay from $2 \times 10^{-4}$ to $5 \times 10^{-4}$. We can see that this seemingly unimportant difference sends the TRADES model back to the state-of-the-art position in the benchmark.

Table 16: We retrieve the results of top-rank methods from https://github.com/fra31/auto-attack. All the methods listed below do not require additional training data on CIFAR-10. Here the model of **Ours (TRADES)** corresponds to lines of weight decay $5 \times 10^{-4}$, eval BN mode and ReLU activation in Table 9, which only differs from the original TRADES in weight decay. We run our methods 5 times with different random seeds, and report the mean and standard deviation.

| Threat model: $\ell_\infty$ constraint, $\epsilon = 8/255$ | | | |
|---|---|---|---|
| Method | Architecture | Clean | AA |
| **Ours (TRADES)** | WRN-34-20 | 86.43 | 54.39 |
| **Ours (TRADES)** | WRN-34-10 | $85.49 \pm 0.24$ | $53.94 \pm 0.10$ |
| Pang et al. (2020c) | WRN-34-20 | 85.14 | 53.74 |
| Zhang et al. (2020) | WRN-34-10 | 84.52 | 53.51 |
| Rice et al. (2020) | WRN-34-20 | 85.34 | 53.42 |
| Qin et al. (2019) | WRN-40-8 | 86.28 | 52.84 |

| Threat model: $\ell_\infty$ constraint, $\epsilon = 0.031$ | | | |
|---|---|---|---|
| Method | Architecture | Clean | AA |
| **Ours (TRADES)** | WRN-34-10 | $85.45 \pm 0.09$ | $54.28 \pm 0.24$ |
| Huang et al. (2020) | WRN-34-10 | 83.48 | 53.34 |
| Zhang et al. (2019b) | WRN-34-10 | 84.92 | 53.08 |

## B.5   MORE EVALUATIONS ON LABEL SMOOTHING

In Table 17 we further investigate the effect of label smoothing on adversarial training.

Table 17: Test accuracy (%) under different **label smoothing** on CIFAR-10. The model is ResNet-18 trained by PGD-AT. We evaluate under PGD-1000 with different number of restarts and step sizes. Here we use the cross-entropy (CE) objective and C&W objective (Carlini & Wagner, 2017a), respectively. We also evaluate under the SPSA attack (Uesato et al., 2018) for $10,000$ iteration steps, with batch size 128, perturbation size 0.001 and learning rate of $1/255$.

| Evaluation method | | | Label smoothing | | | | |
|---|---|---|---|---|---|---|---|
| Attack | Restart | Step size | 0 | 0.1 | 0.2 | 0.3 | 0.4 |
| PGD-1000 (CE objective) | 1 | 2/255 | 52.45 | 52.95 | 53.08 | 53.10 | **53.14** |
| | 5 | 2/255 | 52.41 | 52.89 | 53.01 | **53.04** | 53.03 |
| | 10 | 2/255 | 52.31 | 52.85 | 52.92 | **53.02** | 52.96 |
| | 10 | 0.5/255 | 52.63 | 52.94 | **53.33** | 53.30 | 53.25 |
| PGD-1000 (C&W objective) | 1 | 2/255 | 50.64 | 50.76 | **51.07** | 50.96 | 50.54 |
| | 5 | 2/255 | 50.58 | 50.66 | **50.93** | 50.86 | 50.44 |
| | 10 | 2/255 | 50.55 | 50.59 | **50.90** | 50.85 | 50.44 |
| | 10 | 0.5/255 | 50.63 | 50.73 | 51.03 | **51.04** | 50.52 |
| SPSA-10000 | 1 | 1/255 | 61.69 | 61.92 | **61.93** | 61.79 | 61.53 |

