# OpenReview forum: "Bag of Tricks for Adversarial Training"
_ICLR.cc/2021/Conference — ICLR 2021 Poster_

### Official Review · AnonReviewer3 · 2020-10-27
**An interesting  contribution for identifying good hyperparameters for adversarial training**

**Rating:** 5
**Confidence:** 3

**Review:**

The paper provides an evaluation of different hyperparameter settings for adversarial training. Specifically, it evaluates combinations of warmup, early stopping, weight decay, batch size and other parameters on adversarially trained models. The paper states that its overarching goal is to ``investigate how the implementation details affect the performance of the adversarial trained models''.

Strength
- The papers extensive empirical results are useful to identify good hyperparameters, and it obtains some interesting findings, such that small differences in weight decay can make a big difference in performance, in contrast to standard performance.
- It is interesting to point out, as the paper does, that the robust overfitting phenomena and in particular how to obtain good performance with early stopping depends on a number of hyperparameter setups and is quite sensitive to the particular choice of hyperparameters.

Weaknesses
- The papers novelty is low; essentially it is a rigorous study on how to choose hyperparameters for a specific adversarial training setup (i.e., adversarial training of CIFAR-10).
- One of the papers main contribution is that ``our empirical results suggest that improper  training settings can largely degenerate the model performance while this degeneration may be mistakenly ascribed to the methods themself''. It is well known a given model only performs well if properly trained and if the hyperparameters are chosen appropriately.

The paper's value is in identifying good hyperparameters for adversarial training for CIFAR-10. The paper does a very good job in identifying good hyperparameters and setups, but its main finding that ``we should more carefully fine-tune the training settings in adversarial training'' is common sense and in my opinion not a sufficient contribution for a ICLR publication.

----
UPDATE: Thanks for the response, I have responded below and kept the score constant. Since I'm currently not actively working on the practice of adversarial robustness, the other reviewers are likely a better judge on the usefulness of the results of the paper for the community.

---

> ### Author Response · Authors · 2020-11-14
> **Thank you for your valuable review**
>
> Thank you for your valuable review. We have uploaded a revision of our paper.
>
> ***Question 1: About contributions and novelty***
>
> Here is an example of how our work contributes to the community: Zhang et al. [1] first showed that TRADES outperforms PGD-AT. Later, Rice et al. [2] find that PGD-AT outperforms TRADES (note that TRADES also uses early stopping), which contradicts the results in Zhang et al. Recently, Gowal et al. [3] find that TRADES actually outperforms PGD-AT, which again contradicts the results in Rice et al. Unfortunately, neither Gowal et al. nor Rice et al. detailed on why the contradictions happen, while this paradox could confuse future researches and throw doubts on the correctness of these seminal papers. In this case, our work contributes to a simple answer for this paradox: all the three previous papers report correct results, the key is that they use different values of weight decay.
>
> From the above example, we emphasize that our novelty is to investigate the 'mostly overlooked' hyperparameters, which could become confounders when comparing different defenses. These hyperparameters were usually set by default without carefully tuning, and their effects were underestimated. Our empirical results reveal that adversarial robustness could be much more sensitive to these 'default settings' than we thought, and their effects may override the benefits from the proposed defenses.
>
> ***Question 2: Hyperparameter tuning is common sense***
>
> We totally agree that 'hyperparameter tuning can be important' is a commonly known principle. However, as shown in Table 1, this principle is poorly followed in the community of adversarial training. Compared to standard training, the basic settings used in adversarial training are far from consistent, which means a direct comparison between two defenses could be unfair and may not reflect the effectiveness of the proposed methods. However, this fact is not paid enough attention in the adversarial community before our work. An evidence is that most of the defenses and benchmarks only treat model capacity as the control variable, and leave other settings unconcerned. In our work, we
> comprehensively evaluate the effects of these overlooked hyperparameters, and appeal to the concerns on the unnoticed confounders when benchmarking or evaluating adversarial defenses.

---

### Official Review · AnonReviewer4 · 2020-10-27
**This paper provides a comprehensive evaluations on the different hyper parameter settings of training adversarially robust models. I have a few concerns regarding consistency of the experiments performed here.**

**Rating:** 7
**Confidence:** 4

**Review:**

This paper provides a comprehensive evaluations on the different hyper parameter settings of training adversarially robust models. Consequently, this paper also gives a few takeaways based on the results that they find.

The task of finding good hyper parameters for adversarial training is a challenge. So I like the idea of performing a study on the effects of the different settings.
Since this paper is evaluating the various settings, I have a few concerns regarding consistency of the experiments performed here.

- This is my major concern of this paper.
Most of the results shown in the paper are quantitative. Some qualitative understanding of at least one of the settings---like weight decay, would have been very helpful. For example: Visualization of the loss landscape when changing a hyper parameter as shown in Figure 3 in [1] or visualization of the cross section of the decision boundary as shown in Figure 1 in [2].
Such qualitative evaluations would have helped to also translate these findings across datasets.

- Consistency of experimental settings: Table 2 (change in early stopping/warmup on lr/perturb), Table 5 (change in optimizers) and Table 6  (change in activations) report performance on ResNet-18. On the other hand, Table 8 and Table 9 report findings on WRN-34-10/20.
It would have been more consistent to check Table 2 and Table 5 with WRN-34-10/20 so that the reader finds exactly what happens when the respective changes are made.

- Reportings: In the abstract as well as the introduction, claims are made without giving the context of the settings.
“For example, a slightly different value of weight decay can reduce the model robust accuracy by more than 7%, which is probable to override the potential promotion induced by the proposed methods” and
“In our experiments (e.g., Table 8), we show that the two slightly different settings can differ the robust accuracy by ∼ 4%, which is significant according to the reported benchmarks.”
It would be beneficial to the reader to mention the dataset used here.

- A few minor corrections but not limited to:
“where the learning rate decays at 75 epoch and the training is stopped at 76 epoch.”
to “where the learning rate decays after 75 epochs and the training is stopped after 76 epochs.”

[1] https://arxiv.org/pdf/1807.10272.pdf
[2] https://openaccess.thecvf.com/content_CVPR_2019/papers/Moosavi-Dezfooli_Robustness_via_Curvature_Regularization_and_Vice_Versa_CVPR_2019_paper.pdf

---

> ### Author Response · Authors · 2020-11-14
> **Thank you for the supportive review**
>
> Thank you for the supportive review and kind suggestions. We have uploaded a revision of our paper.
>
> ***Question 1: Qualitative results***
>
> In the revision, we plot the visualization of the cross section of the decision boundary in Figure 3. We can see that proper values of weight decay (e.g., $5\times10^{-4}$) can enlarge margins from decision boundary and improve robustness. Nevertheless, as shown in the left two columns, this effect is less significant on promoting clean accuracy. This is consistent with the learning curve in Figure 1(b).
>
> ***Question 2: Table 2 and Table 5 with WRN-34-10/20***
>
> Thank you for the suggestion, we are running the experiments (will take days to finish). Results will be updated in the revision as soon as possible.
>
> ***Question 3: Clarifications on the dataset and minor corrections***
>
> In the revision, we clarify the dataset CIFAR-10 in abstract and earlier in introduction (second paragraph). We also polish our writing as you suggested.

---

> ### Author Response · Authors · 2020-11-17
> **Updated results**
>
> Thank you for waiting. In the updated revision, the results of Table 2 and Table 5 on WRN-34-10 are reported in Table 14 and Table 15. We can find that the empirical conclusions on WRN-34-10 are consistent with those on ResNet-18.

---

### Official Review · AnonReviewer2 · 2020-10-28
**Great systematic survey - but with limitations that should be addressed or pointed out**

**Rating:** 7
**Confidence:** 4

**Review:**

The paper systematically reviews different hyperparameter settings and training strategies used for PGD adversarial training on CIFAR-10. Based on that, it derives practical takeaways (“bag of tricks”) and puts forward a standard baseline setting for future work.

I think this work is a valuable contribution to the research on adversarial training. Some of the influencing factors that it examines, like early stopping or learning rates, have been occasionally discussed in prior works, others - like weight decay - seem to have been mostly overlooked. The systematic investigation of the effect of all these different factors leads to practical insights and will hopefully allow researchers to measure advances in adversarial training without confounding effects of different hyperparameter settings. In addition, the rich bibliography gives a great survey of the current state-of-the-art.

Having said all that, the paper has several limitations which should be either addressed or clearly pointed out:
- The paper effectively only considers PGD adversarial training on CIFAR-10. In principle this is fine as this is a scenario that has been widely studied and still poses many open research questions. However, I think the limitation to CIFAR-10 should be clearly pointed out in the abstract (currently it’s only mentioned in passing at the end of the introduction), otherwise the percentage values referenced in the abstract are taken out of context, and readers who are interested in adversarial training on larger-scale dataset could be misled (ImageNet is completely outside the scope of this paper). Similarly, the paper does not include some of the recent “free” or “fast” adversarial training protocols in its investigation; again this is fine in principle, and I understand that investigating all the hyperparameters along this additional dimension would dramatically increase the complexity of this study, nevertheless I think it’s a limitation that should be pointed out upfront.
- More details on the attack default settings should be provided. In particular, for PGD: what is the step size, how many (if any) random (re-)starts were performed? I suppose an untargeted PGD attack was used? Did you use the true or predicted labels in the untargeted attacks? Same for AutoAttack: what were the hyperparameters, in particular for AutoPGD? On a related note, was it necessary to run the whole suite of AutoAttack (including FAB and Square)? PGD adversarial training usually isn’t prone to gradient masking, so I would expect that a plain white-box attack like (Auto)PGD should be sufficient for evaluations and lead to more interpretable results. If gradient masking was an issue for some of the hyperparameter settings, this would be an important insight which should be explicitly pointed out. With AutoAttack, unfortunately, such aspects are hidden in a black-box.
- In the takeaways on page 7, I wasn’t sure where (iv) had been discussed previously - could you please point me to it?
- Finally, I’m trying to reconcile the reported results with those from Madry et al. (2017, https://arxiv.org/pdf/1706.06083.pdf) who report 45.8% adversarial accuracy under a PGD attack - which is lower than all except one result reported in this paper (if I’m parsing it correctly). Could you please help me understand the discrepancy or where the datapoint from Madry’s paper should fit in?

A few minor observations:
p.1: “much higher than if in” -> remove “if”
p.1: “These motivate us” -> missing word
p.2: regarding certification methods: calling them “exciting” is a bit casual; also I wouldn’t say that they “cannot match” empirical performance of adversarial training (or do you have a proof?) but rather they “currently do not match” it.
p.3: “with the learning rate decays” -> “decaying”
p.3: “at 75 epoch” and “at 76 epoch” -> “at the 75th epoch” or “at epoch 75” etc.
p.6: “the weight decay in previous work almost falls in three values” -> I think this could be rephrased / explained more precisely

---

> ### Author Response · Authors · 2020-11-14
> **Thank you for the supportive review**
>
> Thank you for the supportive review and kind suggestions. We have uploaded a revision of our paper.
>
> ***Question 1: Clarifications on the dataset and attacking details***
>
> In the revision, we clarify the dataset CIFAR-10 in abstract and earlier in introduction (second paragraph). We detail more on the attack default settings for PGD and AutoAttack in the beginning of Sec.3.
>
> ***Question 2: FastAT and FreeAT***
>
> In the revision, we add evaluations on FastAT and FreeAT in Table 10. The results show that our observations on PGD-AT can generalize well across different AT frameworks.
>
> ***Question 3: Gradient masking***
>
> In the revision, we further evaluate label smoothing under black-box RayS in Table 4, as well as PGD-1000 and SPSA in Table 17. As seen, AutoAttack accuracy is still a good approximation for the worst-case performance. We also modify the discussions on label smoothing, to avoid overclaim as to gradient obfuscation.
>
> ***Question 4: (iv) in takeaways***
>
> This is observed in Table 2 (as well as Table 14 in the revision). The AutoAttack accuracy for baseline is $48.51\%$, and for early stopping attack iter. is around $46\%$.
>
> ***Question 5: Discrepancy from the results in Madry et al.***
>
> In the training phase, we apply PGD-10, weight decay $5\times10^{-4}$, and early stopping training epochs; Madry et al. apply PGD-7, weight decay $2\times10^{-4}$, and without early stopping. In the evaluation phase, we test under PGD-10, while Madry et al. test under PGD-20.
>
> ***Question 6: Minor observations***
>
> We appreciate for the kind suggestions, and we have modified them in the revision.

---

### Official Review · AnonReviewer1 · 2020-10-28
**The paper lacks novelty, although the observations are insightful**

**Rating:** 6
**Confidence:** 5

**Review:**

################################ Summary ###################################

The authors investigate the impact of various training hyperparameters such as weight-decay, batch size, use of batch normalization in eval/ train mode during adversary generation, smooth activations, optimizer and learning rate schedule. The authors demonstrate that use of the right hyperparameters can bring TRADES back to the top of the AutoAttack leaderboard. (~1.6% boost)

################################# Pros ######################################

  -  The paper presents a good review of the tricks used in different adversarial defense papers, and presents the impact of varying each of them on adversarial robustness and clean accuracy
  -  The authors highlight the importance of using the right training hyperparameters across different baselines for a fair comparison.
  -  Based on experiments with the PGD-AT model, the authors recommend a set of hyperparameter settings which can potentially generalize to other defenses.
  -  The authors show that although the proposed settings were found on the PGD-AT model, they generalize well to the TRADES defense as well.

################################# Cons ######################################

  -  The paper lacks novelty, since it merely presents results of existing defenses with different hyperparameters. An analysis/ explanation of why these settings matter more for adversarial training could add value to the submission.
  -  It is a well known fact that these hyperparameter settings are important for the standard training of Deep Networks. The work by Rice et al. [1] showed that early stopping is important for better adversarial robustness. They also showed the impact of varying settings such as learning rate schedule, weight decay and other regularizers in the supplementary section. This already highlights the importance of selecting the right hyperparameters for adversarial training. Hence, the finding in this paper is not too surprising.
  -  All the results reported in the paper are for single runs, however they may be a result of variance due to random initialization. Reporting statistics (such as mean and variance) across multiple reruns would be more helpful.
  -  The final hyperparameter settings suggested in the paper are very similar to those used by Rice et al., with the exception of using Batch normalization in eval mode during adversary generation. Also, the impact of train / eval mode during Batch norm is not consistent across Tables - 7, 8 and 9. Train mode seems to be better for large models on PGD-AT, whereas the eval mode is better for TRADES.
  -  While the authors show that the hyperparameter settings found on PGD defense generalize well to TRADES, it would have been very useful to show its impact on some of the other defenses listed in Table-1 as well. This would highlight whether the use of a common set of hyperparameters across baselines must be encouraged, or whether it is better to stick to the implementation of the respective authors.
  -  The paper mentions the use of AutoAttack to evaluate the impact of label smoothing in order to rule out gradient obfuscation. However, AutoAttack does not completely rule out the possibility of gradient masking in a defense [2]. The impact of label smoothing on adversarial robustness is still debatable [3, 4]. The defense by Pang et al. [5] is also broken by Tramer et al. [6].

[1] Rice et al., Overfitting in adversarially robust deep learning, ICML 2020, https://arxiv.org/abs/2002.11569

[2] Croce et al., RobustBench: a standardized adversarial robustness benchmark, https://arxiv.org/pdf/2010.09670.pdf

[3] https://openreview.net/forum?id=Bylj6oC5K7

[4] https://openreview.net/forum?id=BJlr0j0ctX

[5] Pang et al., Improving adversarial robustness via promoting ensemble diversity, ICML 2019.

[6] Tramer et al., On Adaptive Attacks to Adversarial Example Defenses, NeurIPS 2020, https://arxiv.org/pdf/2002.08347.pdf

############################### Reasons for score ##############################

Although the paper is an interesting read and guide for training adversarial defenses, the key finding of the paper (i.e., hyperparameter tuning can be important for adversarial robustness) is not new to the community, and has been highlighted in recent work [1]. Hence I vote to reject the paper.

##############  Additional Feedback (not part of decision assessment) #####################

  -  Could the authors clarify whether a validation split has been used for early stopping of TRADES in Table-13, and in general for other defenses reported throughout the paper? If yes, what is the size of the validation split? Also, what is the criteria used for early stopping?
  -  As discussed in the paper, all other defenses in Table-13 also need to be rerun using the optimal settings for a fair comparison.
  -  It would be useful to include WideResNet architecture with different capacities in Fig.2 of the paper, since this is a common architecture choice across multiple defenses.

##############  Update after rebuttal ##############

I would like to update the score to 6 based on the author's response. I have not increased further due to the limited novelty of the paper. However, the observations in the paper certainly add value to the research community.

I request the authors to consider reporting performance of other defenses in Table-16 using the recommended settings in their final version.

---

> ### Author Response · Authors · 2020-11-14
> **Thank you for your valuable review (Part 2/2)**
>
> ***Question 4: Analysis of why these settings matter more for adversarial training***
>
> In the revision, we plot the visualization of the cross sections of the decision boundary in Figure 3. We can see that proper values of weight decay (e.g., $5\times10^{-4}$) can enlarge margins from decision boundary and improve robustness. Nevertheless, as shown in the left two columns, this effect is less significant on promoting clean accuracy. This is consistent with the learning curve in Figure 1(b).
>
> Besides, we did analyze the mechanisms of several critical settings in our paper, based on related analyses in previous work. For examples, we find that moderate label smoothing is beneficial, which can be regarded as the effect induced by calibrating the confidence [4]; we find that residual connections can promote model performance, which can be explained as easier generation of highly transferable adversarial examples during adversarial training [5]; we find that train BN mode has worse clean accuracy than eval BN model, which can be viewed as blurring the recorded distribution [6].
>
> Meanwhile, we also leave some of the conclusions (e.g., warmup, optimizer, etc.) as purely empirical, to avoid providing ambiguous explanations that may mislead other researchers.
>
> ***Question 5: Multiple runs and report statistics***
>
> In the revision, we rerun our results in Table 13 (Table 16 in the revision) on TRADES and report the statistics. Although running all the experiments for multiple times is too expensive, we will figure out some critical experiments and provide statistics to better support our conclusions in later revisions.
>
> ***Question 6: Experiments on other defenses***
>
> In the revision, we add evaluations on FastAT and FreeAT in Table 10. The results show that our observations on PGD-AT can generalize well across different AT frameworks. Besides, we do not advocate that all the defenses must follow the baseline settings. Instead, we highlight that many default settings used by previous implementations could be not appropriate. If there needs a certain default choice, then our baseline would generally work better (not always).
>
> ***Question 7: About experiments***
>
> The eval BN mode consistently performs better than train BN mode on clean accuracy, while keeping comparable robust accuracy. In the revision, we further evaluate label smoothing under black-box RayS in Table 4. As seen, AutoAttack accuracy is still a good approximation for the worst-case performance. We also modify the discussions on label smoothing, to avoid overclaim as to gradient obfuscation. Besides, we did not claim that label smoothing can prevent the model from evaded by adaptive attacks. Our conclusion is that moderate label smoothing can be beneficial for adversarial training.
>
> ***Question 8: Suggested setting***
>
> Rice et al. mainly advocate early stopping the training epoch, while they did not explicitly justify their training settings. An evidence is that many papers after Rice et al. still follow the training setting from Zhang et al. or Madry et al. [7]. In contrast, we provide the basic settings after comprehensively evaluating the effects.
>
> ***Question 9: Additional feedback***
>
> We do not split a validation set, and we apply the best-performing model selection principle, which is justified in the last paragraph of Sec.3.2 of Rice et al. In Fig.2, we choose the architectures with comparable model size (i.e., number of parameters). We could include WRNs in a wider plot.
>
> Reference:
>
> [1] Zhang et al. Theoretically Principled Trade-off between Robustness and Accuracy, ICML 2019, https://arxiv.org/abs/1901.08573
>
> [2] Rice et al., Overfitting in Adversarially Robust Deep Learning, ICML 2020, https://arxiv.org/abs/2002.11569
>
> [3] Gowal et al. Uncovering the Limits of Adversarial Training against Norm-Bounded Adversarial Examples, https://arxiv.org/abs/2010.03593
>
> [4] Stutz et al. Confidence-calibrated Adversarial Training: Generalizing to Unseen Attacks, ICML 2020, https://arxiv.org/abs/1910.06259
>
> [5] Wu et al. Skip Connections Matter:On the Transferability of Adversarial Examples Generated with Resnets, ICLR 2020, https://openreview.net/forum?id=BJlRs34Fvr
>
> [6] Xie et al. Intriguing Properties of Adversarial Training at Scale, ICLR 2020, https://arxiv.org/abs/1906.03787
>
> [7] Madry et al, Towards deep learning models resistant to adversarial attacks, ICLR 2018, https://arxiv.org/abs/1706.06083

---

> > ### Comment · AnonReviewer1 · 2020-11-18
> > **Thank you for the detailed response, label smoothing experiments need better evaluation**
> >
> > I would like to thank the authors for the detailed response. This addresses most of my concerns.
> >
> > Although label smoothing is not the primary contribution of the paper, I believe it may be misleading to include these results without extensive evaluation. The papers that have been cited for initial work on label smoothing have been invalidated subsequently [1, 2]. Also the results in [3] show clearly that label smoothing is susceptible to attacks with larger number of steps. Since label smoothing is known for its gradient masking effects, it is important to do a thorough evaluation as suggested by Carlini et al. [4]. The fact that RayS was weaker does not prove that Autoattack evaluation is sufficient. Although the authors suggest to use a mild extent of label smoothing, there is no evidence in prior work that this should help. If label smoothing cannot individually help, it is likely that even a mild form of it combined with adversarial training will not help. The results related to label smoothing may therefore be misleading to the research community. I suggest the authors to do a more comprehensive evaluation and modify the discussion on label smoothing based on their findings. Some of the attacks may include (not an extensive list) : PGD attacks with CE and max-margin based loss (CW loss [6]) with larger number of steps (~1000) and restarts (until the results indicate that the attack has converged), B&B l-inf attack [5], Multi-Targeted attack [7], SPSA attack [8].
> >
> > [1] https://openreview.net/forum?id=BJlr0j0ctX
> >
> > [2] Pang et al., Improving adversarial robustness via promoting ensemble diversity, ICML 2019.
> >
> > [3] https://openreview.net/forum?id=Bylj6oC5K7
> >
> > [4] Carlini et al., On evaluating Adversarial Robustness, https://arxiv.org/abs/1902.06705
> >
> > [5] Brendel et al,. Accurate, reliable and fast robustness evaluation
> >
> > [6] Carlini et al., Towards evaluating the robustness of neural networks
> >
> > [7] Gowal et al., An Alternative Surrogate Loss for PGD-based Adversarial Testing, https://arxiv.org/pdf/1910.09338.pdf
> >
> > [8] Uesato et al., Adversarial risk and the dangers of evaluating against weak attacks

---

> > > ### Author Response · Authors · 2020-11-19
> > > **More evaluations on label smoothing are updated**
> > >
> > > Thank you for the suggestions. We update the revision which further includes:
> > >
> > > 1. In Table 17, we evaluate the effect of label smoothing under PGD-1000 and SPSA. For PGD-1000, we apply the cross-entropy (CE) objective and C&W objective, respectively. We report the results under the different number of restarts and step sizes. For SPSA, we run for 10,000 iteration steps, with batch size 128, perturbation size 0.001, and step size 1/255. We can observe that at least under the attacks that we evaluated, mild label smoothing (e.g., LS$=0.2\sim0.3$) indeed help adversarial training, whereas excessive label smoothing (e.g., LS$>0.4$) would degrade robustness.
> > >
> > > 2. As you suggested, we modify the discussion on label smoothing in Sec. 3.2. We clarify that combining label smoothing on standard models cannot prevent the models from being evaded by adaptive attacks or a larger number of iterations. We further suggest future researchers to carefully evaluate their proposed defenses when applying the trick of label smoothing.

---

> ### Author Response · Authors · 2020-11-14
> **Thank you for your valuable review (Part 1/2)**
>
> Thank you for your valuable review. We have uploaded a revision of our paper.
>
> ***Question 1: About contributions and novelty***
>
> Here is an example of how our work contributes to the community: Zhang et al. [1] first showed that TRADES outperforms PGD-AT. Later, Rice et al. [2] find that PGD-AT outperforms TRADES (note that TRADES also uses early stopping), which contradicts the results in Zhang et al. Recently, Gowal et al. [3] find that TRADES actually outperforms PGD-AT, which again contradicts the results in Rice et al. Unfortunately, neither Gowal et al. nor Rice et al. detailed on why the contradictions happen, while this paradox could confuse future researches and throw doubts on the correctness of these seminal papers. In this case, our work contributes to a simple answer for this paradox: all the three previous papers report correct results, the key is that they use different values of weight decay.
>
> From the above example, we emphasize that our novelty is to investigate the 'mostly overlooked' hyperparameters, which could become confounders when comparing different defenses. These hyperparameters were usually set by default without carefully tuning, and their effects were underestimated. Our empirical results reveal that adversarial robustness could be much more sensitive to these 'default settings' than we thought, and their effects may override the benefits from the proposed defenses.
>
> ***Question 2: Differences from Rice et al.***
>
> The settings that Rice et al. evaluated: early stopping (w.r.t. training epoch), learning rate schedule (e.g., piecewise, cyclic, etc.), $l_1$ and $l_2$ regularization, cutout, mixup, semi-supervision.
>
> The settings that we evaluated:  early stopping (w.r.t. adversarial intensity), warmup, batch size, label smoothing, optimizer, weight decay, activation function, model architecture, BN mode.
>
> The only overlapping between Rice et al. and ours is weight decay (i.e., $l_2$ regularization). As we already clarified in footnote 3 (footnote 4 in the revision), Rice et al. focus on a coarse value range of $\{5\times 10^{k}\}$, where $k\in\{-4,-3,-2,-1,0\}$, in order to test the effect of $l_2$ regularization on overfitting. In contrast, we choose the values of $1\times 10^{-4}$, $2\times 10^{-4}$, and $5\times 10^{-4}$ based on the implementations details of previous defenses, and investigate the effect of slightly different weight decay (the difference is usually overlooked) on model robustness.
>
> ***Question 3: Hyperparameter tuning is common sense***
>
> We totally agree that 'hyperparameter tuning can be important' is a commonly known principle. However, as shown in Table 1, this principle is poorly followed in the community of adversarial training. Compared to standard training, the basic settings used in adversarial training are far from consistent, which means a direct comparison between two defenses could be unfair and may not reflect the effectiveness of the proposed methods. However, this fact is not paid enough attention in the adversarial community before our work. An evidence is that most of the defenses and benchmarks only treat model capacity as the control variable, and leave other settings unconcerned. In our work, we
> comprehensively evaluate the effects of these overlooked hyperparameters, and appeal to the concerns on the unnoticed confounders when benchmarking or evaluating adversarial defenses.

---

> ### Author Response · Authors · 2020-11-19
> **Thank you again**
>
> Thank you for updating the score, and your feedback really helped us on improving our work. We will further polish our paper with complete results in the final version.

---

### Decision · Program_Chairs · 2021-01-07
**Final Decision**

**Decision:**

Accept (Poster)

**Comment:**

The authors have conducted a thorough empirical study on the hyperparameters of representative adversarial training methods. The technical novelty of this paper might be insufficient.  But the empirical findings in this paper explain the strange and inconsistent reported algorithm results in the literature to some extent and remind the necessity and importance of a careful study on hyperparameters. The authors have actively interacted with the reviewers and through the discussions, many unclear issues have been fixed.